# Correction of a widespread bias in pooled chemical genomics screens improves their interpretability

Lili M Kim [ID][1,4], Horia Todor [ID][1,4 ✉] & Carol A Gross[1,2,3]

## Abstract

**Chemical genomics is a powerful and increasingly accessible technique to probe gene function, gene–gene interactions, and antibiotic synergies and antagonisms. Indeed, multiple large-scale pooled datasets in diverse organisms have been published. Here, we identify an artifact arising from uncorrected differences in the number of cell doublings between experiments within such datasets. We demonstrate that this artifact is widespread, show how it causes spurious gene–gene and drug–drug correlations, and present a simple but effective post hoc method for removing its effects. Using several published datasets, we demonstrate that this correction removes spurious correlations between genes and conditions, improving data interpretability and revealing new biological insights. Finally, we determine experimental factors that predispose a dataset for this artifact and suggest a set of experimental and computational guidelines for performing pooled chemical genomics experiments that will maximize the potential of this powerful technique.**

**Keywords** Chemical Genomics; Pooled Screens; Data Analysis; Bacterial Screens; Error Correction
**Subject Categories** Biotechnology & Synthetic Biology; Computational Biology; Methods & Resources

## Introduction

Understanding the function of bacterial genes in diverse environments is an outstanding problem with major implications for antibiotic discovery and use. Solving this problem has become increasingly important as our understanding of the role of bacteria in human health, disease, and industry continues to grow. Chemical genomics, a technique in which the fitness of mutants in genome-wide loss-of-function libraries is assayed under multiple drug or environmental conditions, has emerged as a central strategy for addressing this problem. Chemical genomics allows high-throughput discovery of phenotypes for uncharacterized genes (Giaever et al, 2002; Nichols et al, 2011; Price et al, 2018),

gene–gene interactions (Typas et al, 2010; Sher et al, 2020), and antibiotic mechanisms of action (Peters et al, 2016; Santiago et al, 2018; Sher et al, 2020; Li et al, 2022). The advent of pooled sequencing-based approaches such as Tn-seq (van Opijnen et al, 2009; Wetmore et al, 2015) and variations of CRISPRi-seq (Qi et al, 2013; Hawkins et al, 2020; Todor et al, 2021; Banta et al, 2024; Dénéréaz et al, 2024; Koo et al, 2024) combined with the decrease in DNA sequencing and synthesis costs has greatly accelerated these experiments. Pooled, high-throughput chemical genomics datasets have now been published for dozens of bacterial species (Nichols et al, 2011; Santiago et al, 2018; Price et al, 2018; Sher et al, 2020; Li et al, 2022; Ward et al, 2024).

A pooled growth experiment estimates the fitness of each mutant strain in a library by comparing its relative abundance (measured via sequencing) before and after growth in a condition of interest. Changes in the relative abundance of each strain are quantified as log2-fold change (L2FC) and reflect strain fitness. Strains growing slower than wild-type will be diluted by the overall growth of the library, resulting in a decrease in their relative abundance, while strains growing faster than wild-type will outcompete the library and increase in their relative abundance. Chemical genomics datasets consist of many such pooled growth experiments performed in the presence of various stresses. To identify genes whose depletion/deletion specifically sensitizes or protects the bacterium from a drug or environmental condition, the reference (no-stress) fitness (or L2FC) of each mutant is typically compared to their fitness in a stress condition to determine their condition-specific change in fitness. This condition-specific change in fitness is expected to be independent of a strain's absolute (or reference) fitness; however, this relies on the assumption that the reference and test experiment have been grown for the same number of cell doublings.

Here, we demonstrate that this assumption is violated in many pooled chemical experiments, resulting in spurious experiment-specific correlations between the reference (no-stress) fitness of strains and their condition-specific change in fitness. We show that this artifact decreases the utility of chemical genomics. We present a simple post hoc method for correcting this artifact that requires knowledge only of the reference fitness of each strain and is broadly applicable to multiple experimental modalities (CRISPRi, Tn-seq, etc) and analysis pipelines (DESeq2, edgeR, etc.). Finally, we show that applying this correction to CRISPRi and Tn-seq chemical genomics datasets improves their biological interpretability and reveals new biology.

[1]Department of Microbiology and Immunology, University of California, San Francisco, San Francisco, CA, USA. [2]Department of Cell and Tissue Biology, University of California, San Francisco, San Francisco, CA, USA. [3]California Institute of Quantitative Biology, University of California, San Francisco, San Francisco 94158 CA, USA. [4]These authors contributed equally: Lili M Kim, Horia Todor. ✉E-mail: horia.todor@gmail.com

# Results and discussion

## Differences in the actual number of strain doublings predictably affect relative strain abundance

The change in the relative abundance of a strain in a pooled growth experiment is a product of its growth rate compared to wild-type and the number of library doublings between the start and end of the experiment: a growth-deficient mutant will be more depleted after 12 doublings of competition than after 10 doublings (Fig. 1A). Condition-specific changes in fitness are calculated by comparing the reference L2FC to the test condition L2FC. Therefore, when there is an uncorrected difference in the number of cell doublings between a reference and test experiment, strains that affect growth will exhibit nonzero condition-specific changes in fitness. For example, if the test (e.g., drug) experiment has fewer cell doublings than the reference experiment, all growth-deficient strains would appear protective, as they would be relatively less depleted in the test condition relative to the reference condition (positive condition-specific changes in fitness; Fig. 1B, purple lines). Conversely, if the test (e.g., drug) experiment has more cell doublings than the reference experiment, all growth-deficient strains would appear sensitizing, as they would be relatively more depleted in the test condition relative to the reference condition (negative condition-specific changes in fitness; Fig. 1B, red lines). Since these effects linearly scale with the reference fitness of each mutant strain, uncorrected differences in the number of cell doublings between experiments manifest as a linear relationship

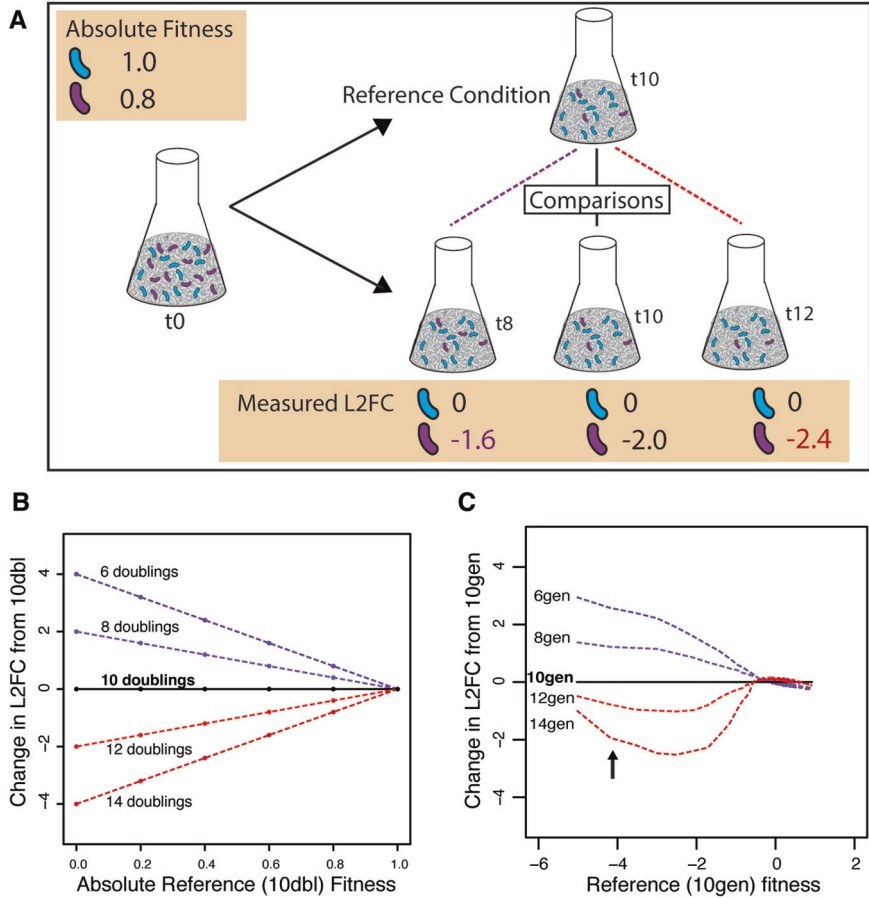

**Figure 1. Number of library doublings affects measured L2FC and condition-specific changes in fitness.**

(A) The measured L2FC of a given strain is dependent on the absolute reference fitness of the mutant and the number of doublings of the library from the start of the experiment to the end. Strains with a growth defect (e.g., knockdown of an essential gene) will deplete from the population with each doubling, independent of an interaction with the test condition (drug, stressor, etc). Therefore, if a reference sample (here, grown for 10 doublings) is compared to a test sample collected too soon (e.g., 8 doublings), growth-deficient strains will appear to have a protective interaction because they are less depleted in the test condition. Conversely, if the same reference sample is compared to a test sample collected too late (e.g., 12 doublings), growth-deficient strains will appear to have a sensitizing interaction because they are more depleted in the test condition. The L2FC of strains with no growth defect (such as CRISPRi strains with non-targeting sgRNAs) does not change as a function of cell doublings. (B, C) In experiments with uncorrected differences in the number of cell doublings, condition-specific change in L2FC of growth-deficient strains will vary as a function of their reference fitness, causing growth-deficient strains to appear protective or sensitizing. The relationship between reference fitness and the condition-specific changes in fitness is approximately linear, and can be either positive (if the reference experiment was shorter than the test experiment) or negative (if the reference experiment was longer than the test experiment). Figures show condition-specific changes in L2FC of theoretical (B) and experimental data (C) normalized to a 10-doubling reference L2FC. Experimental time course data is from (Rishi et al, 2020). Strains with very low reference fitness (arrow) diverge from the linear trend when the test experiment has more doublings than the reference experiment because they become completely diluted from the pool (i.e., read counts = 0).

between reference fitness and condition-specific changes in fitness both theoretically (Fig. 1B) and using data from an experimental time course (Rishi et al, 2020) (Fig. 1C).

Control strains such as those containing non-targeting CRISPRi sgRNAs or transposon insertions in non-essential genes are commonly used to correct for certain biases in pooled screens (Li et al, 2014). However, these control strains cannot be used to correct for differences in the number of cell doublings because the abundance of such strains does not change during the course of an experiment (i.e., absolute fitness = 1.0, Fig. 1A). To demonstrate this, we generated deliberately biased datasets using the data from a *Mycobacterium tuberculosis* CRISPRi chemical genomics screen (Li et al, 2022). For all drugs, we compared each timepoint (1, 5, or 10 days predepletion) to a control with fewer, the same, or more cell doublings ("Methods", Table EV1). As expected, the incorrect comparisons resulted in strong positive or negative correlations between reference and condition-specific change in fitness that depended on whether the control had more or fewer cell doublings than the experimental sample (Fig. EV1A). Despite the strong biases caused by performing these deliberately incorrect comparisons, the median of the non-targeting controls remained near zero (correct control = 0.00, control with fewer doublings = 0.16, control with more doublings = −0.16; Fig. EV1B; Table EV1). To the best of our knowledge, the effect of different numbers of cell doublings is not considered in any widely used analysis pipelines (e.g., edgeR, DEseq, TRANSIT, MAGeCK, DrugZ).

## Differences in the number of cell doublings between experiments are pervasive in pooled chemical genomics screens

Differences in the number of library doublings between samples can arise from several sources. The most common is a lack of careful quantification of bacterial growth over the course of an experiment. Instead, strains are grown for a predetermined number of hours (e.g., Li et al, 2022), or to saturation (e.g., Ward et al, 2024). However, even when bacterial growth is quantified, increased (non-specific) bacterial death or drug-induced morphological changes could result in more or fewer doublings than that calculated from optical density measurements. To determine if this practice results in differences in the number of cell doublings between experiments, we analyzed several previously published datasets for correlations between reference and condition-specific changes in fitness (Li et al, 2022; Price et al, 2018; Sher et al, 2020; Santiago et al, 2018; Ward et al, 2024). We considered both CRISPRi, Tn-seq, and single-gene knockout experiments in diverse bacteria. For each organism, we binned all genes/sgRNAs based on their reference fitness and assessed the correlation between the reference fitness and condition-specific changes in fitness across these bins for each condition ("Methods"). In almost all datasets, we found significant relationships between reference fitness and condition-specific changes in fitness in many experiments (Fig. 2A–F), both positive and negative, suggesting uncorrected differences in the number of cell doublings between experiments. In several cases, we observed both positive and negative relationships between reference fitness and condition-specific changes in fitness for different concentrations of the same drug (e.g., Fig. 2A, purple lines), and we found no conditions or drugs that consistently caused a high or low number of cell doublings (Fig. EV2). Together,

these data suggest that these correlations are the result of experimental artifacts, not biological phenomena such as growth-rate dependent antibiotic susceptibility (Lee et al, 2018).

## Correcting for the number of cell doublings removes spurious correlations between genes and conditions

In many of the datasets and conditions analyzed, reference fitness only mildly affected condition-specific changes in fitness, suggesting that the number of cell doublings in the reference and test experiments were similar though not identical. To determine how this artifact affects downstream analyses, we first developed a method for removing these biases ("Methods"). Briefly, our method bins all the strains in an experiment by their reference fitness and then normalizes the median condition-specific changes in fitness in each bin to zero (Fig. 3, "Methods"). This correction is simple to apply and requires knowledge only of each strain's reference fitness in addition to the condition-specific change in fitness. We applied this correction to one CRISPRi dataset (Li et al, 2022) and one Tn-seq dataset (*Caulobacter crescentus*; Price et al, 2018) and assessed how this correction affected the biological interpretation of the data.

Correlations between the phenotypes of two genes across conditions are indicative of functional connection between genes (Nichols et al, 2011). For both datasets, applying our correction drastically reduced the number of strong positive correlations between growth-deficient strains (Fig. 4A,B), suggesting that many of these correlations are not biologically meaningful but are instead caused by correlated changes in condition-specific changes in fitness due to differences in the number of cell doublings in each experiment. Consistent with this idea, our correction also shifted the median correlation from >0.5 to ~0, as expected if most gene pairs do not interact. To determine if our correction improved the biological interpretability of the data, we assessed the number of strong correlations between growth-deficient genes within and across COG categories in the CRISPRi dataset, which contains phenotype data for most essential genes. Correcting for the number of cell doublings reduced the total number of strong (Pearson's $r > 0.7$) interactions between growth-deficient strains ~tenfold (from 158,642 to 16,350), while reducing the number of strong interactions between strains in the same COG categories significantly less (~fourfold, from 18,687 to 4306, $P < 2.2 \times 10^{-16}$, $\chi^2$ test), suggesting that our correction preferentially removed spurious correlations between unrelated growth-deficient strains (Fig. 4C,D). Similar trends were observed in the *C. crescentus* Tn-seq data. These trends are apparent in individual examples. For example, without correction the phenotypes of the *C. crescentus lexA* (the central regulator of DNA repair) strain are correlated ($r > 0.5$) to those of 651 strains, of which only 11 (1.7%) are in the "L" (DNA Replication and repair) COG category—a very low rate of biologically meaningful hits (Fig. EV3A). After correction, *C. crescentus lexA* is correlated ($r > 0.5$) to 32 genes, of which 14 (44%) are in the "L" COG category: a much more specific and biologically meaningful result (Fig. EV3B,C). Our correction affects genes proportionately to their growth defect (or growth advantage), and therefore most strongly affects the condition-specific changes in fitness of essential genes. Since understanding the relationship between essential gene–gene and gene-drug interactions is important for determining antibiotic targets as well as for understanding bacterial biology, our correction plays an important role in interpreting these data.

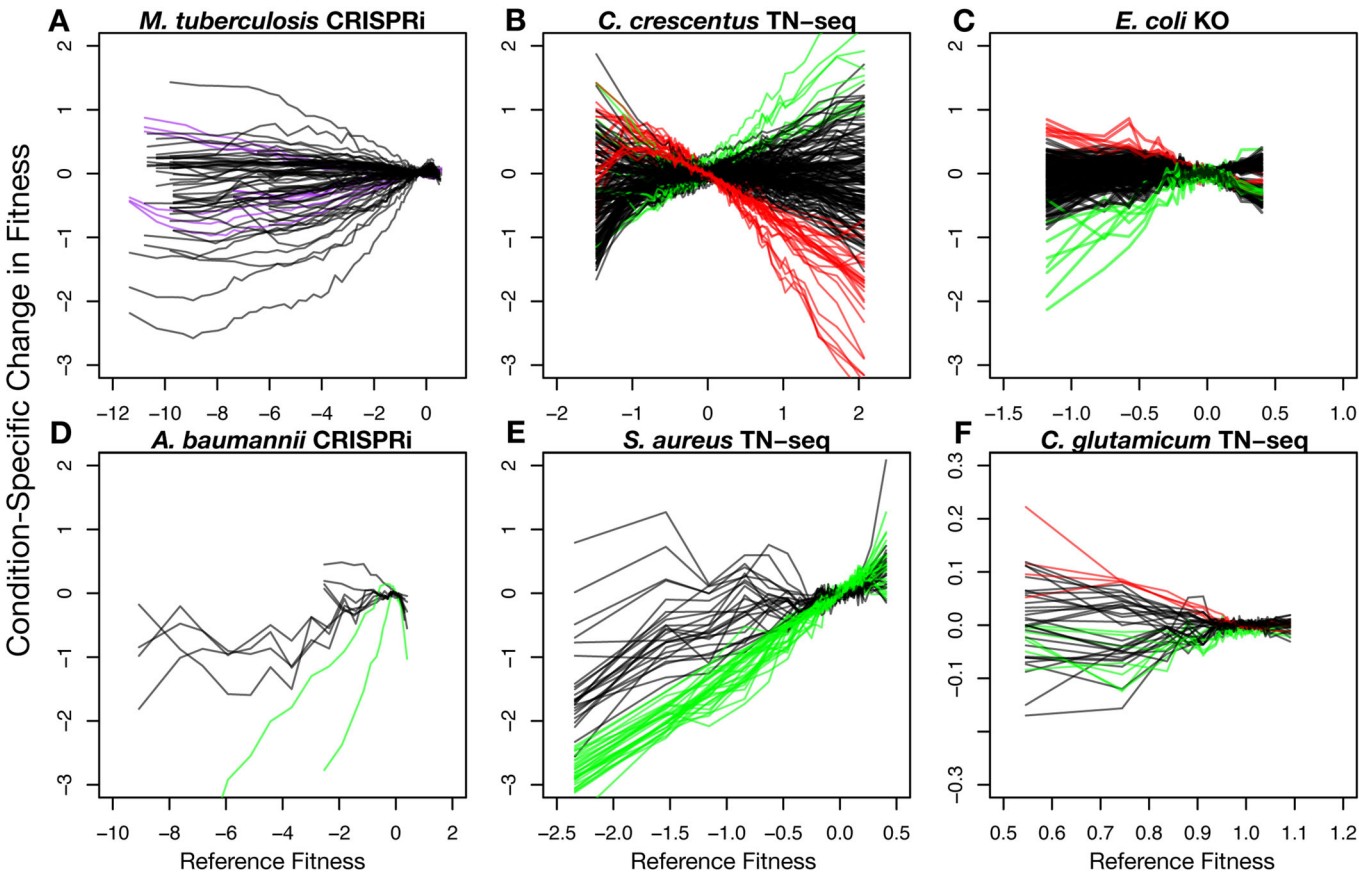

**Figure 2. Experimental length varies within published CRISPRi, Tn-seq, and single-gene deletion (KO) datasets across diverse bacteria.**

Each panel depicts the relationship between reference and condition-specific changes in fitness for each experiment in a chemical genomics dataset. Median values of reference and condition-specific changes in fitness are calculated on a per bin basis as described ("Methods") and plotted. (A) *M. tuberculosis* CRISPRi: purple lines denote clarithromycin experiments, some of which have positive relationships and some of which have negative relationships between condition-specific changes in L2FC and reference fitness. (B–F) *C. crescentus* Tn-seq, *E. coli* KO, *A. baumannii* CRISPRi, *S. aureus* Tn-seq, and *C. glutamicum* Tn-seq, respectively. Many experiments exhibited significant correlations between relative and condition-specific changes in fitness (i.e., the line is not flat). Experiments with exceptionally positive or negative slopes are highlighted in green and red (respectively).

Correlations (or anticorrelations) between drugs across strains can indicate potential drug synergies/antagonisms and have been used to ascertain the mechanism of action for novel drugs (Santiago et al, 2018). We reasoned that failure to correct the number of library doublings may also result in spurious correlations and anticorrelations between conditions, complicating these goals. To test this, we compared the correlation between drugs/conditions in the CRISPRi and Tn-seq datasets described above. Though most drug–drug correlations in the CRISPRi data were minimally affected by the correction, we observed a strong correlation between some clarithromycin and vancomycin experiments in the uncorrected data (Fig. 5A,B). Applying our correction abolished this correlation which is unlikely to be biologically meaningful: clarithromycin and vancomycin are unrelated drugs with different targets, and the correlation was only observed for a subset of the clarithromycin samples (Fig. 5A,B). It also abolished other likely spurious correlations (Fig. EV4A). By contrast, the *C. crescentus* Tn-seq data exhibited many more spurious correlations between conditions. To illustrate this, we plotted the correlations between all 110 stress experiments ordered by their estimated number of

library doublings (Fig. 5C,D). In the uncorrected data, stress conditions with fewer (top left) or more (bottom right) number of doublings exhibited strong correlations regardless of the identity of the stressor and were anti-correlated to each other (Fig. 5C). Correcting for the number of doublings removed these biases (Fig. 5D), and drastically lowered the total number of strong ($|r| > 0.7$) correlations/anticorrelations between conditions (from 29 to 11 pairs, Fig. EV4B). Taken together, our analysis strongly suggests that correcting for differences in the number of library doublings has the potential to greatly improve the interpretability of pooled chemical genomics datasets.

## Factors determining the importance of this correction

To understand the factors that determine the importance of our correction, we applied our correction to the 32 identically analyzed bacterial Tn-seq datasets (Price et al, 2018) and evaluated the resulting gene–gene phenotype correlation matrices. The importance of the correction varied (Fig. 6A). Out of 42 datasets with at least 10 experiments, our correction decreased the number of strong ($r > 0.7$)

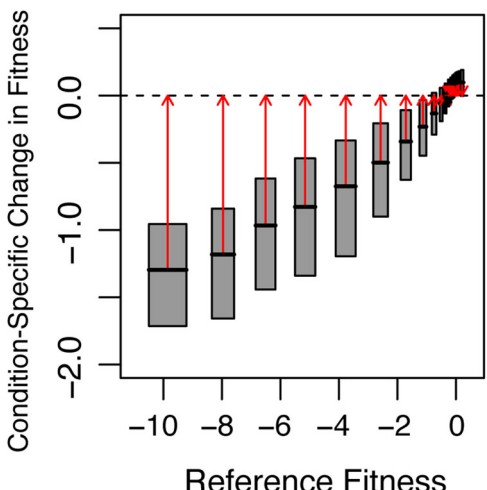

**Figure 3. Schematic of post hoc correction method.**

Mutants are sorted by reference fitness and binned ("Methods"). For each bin (box), the median condition-specific change in fitness (thick black bar in the center of the box) is subtracted from the condition-specific change in fitness of each individual sgRNA/gene (depicted by the red arrow), bringing the median condition-specific change in fitness of the bin to 0 (dashed line). Box height corresponds to the interquartile range (IQR) of the condition-specific changes in fitnesses in that bin, and box width corresponds to the IQR of the reference fitnesses in that bin. The increasing number of bins around Reference Fitness = 1 reflects the greater proportion of gene knockdowns with no fitness defect. Data shown is from the *M. tuberculosis* CRISPRi dataset, Day 1 vancomycin 0.0625 μg/mL, with bin size = 5000; total number of bins = 19.

correlations between genes drastically (>tenfold) in three organisms and more modestly (1.5- to 10-fold) in an additional seven bacteria. Similarly, when considering the top ten most correlated genes for each gene, we found that they were quite different (<40% conserved) for three organisms and somewhat different (40–60% conserved) for an additional 9 bacteria. Applying our correction did not introduce additional correlations: only 2 datasets had negligible increases (~1%) in the number of strong correlations after applying the correction. To determine whether the correction improved the biological mean-ingfulness of the data, we assessed how well the corrected and uncorrected correlations captured known interactions between genes in the STRING database in the three organisms in which our correction made the most difference. The corrected data generally better captures known interactions (Fig. EV5). This effect is due primarily to a decrease in spurious correlations—relatively few correlations are increased by our correction. In total, our analysis suggests that correcting for the number of cell doublings improves the biological interpretability of the data by removing spurious correlations.

A major factor driving the effect of our correction on a given dataset was the number of mutant strains with overall higher or lower fitness (+/− 1 L2FC) than the controls (Fig. 6B). Since the measured fitness of such strains are most affected by differences in number of library doublings, the preponderance of these strains in a library affects the importance of our correction. This is demonstrated in the published analysis of the *C. crescentus* data (Price et al, 2018) which found no improvement in data quality when accounting for library doublings because only 7 gene pairs with reference L2FC < −1 were considered in their analysis of 2430/~4,000,000 total gene pairs. Another major factor in the importance of the correction is the

proportion of experiments with many more or fewer doublings than the reference, which require a large correction (Fig. 6C). Together, these data suggest that our correction will be most meaningful for pooled chemical genomics datasets generated without careful OD measurements and those using CRISPRi (which increases the number of strains with strong phenotypes). Such experiments are becoming increasingly attainable due to the decrease in the cost of DNA synthesis and sequencing.

## Summary

High-throughput sequencing-based chemical genetics approaches have greatly expanded our understanding of bacterial gene function in diverse environments. However, much of the power and promise of chemical genomics comes not from the analysis of single gene-condition interactions but from considering phenotypes across drugs and genes. Such analyses have the potential to uncover new associations between genes as well as antibiotic synergies and antagonisms, but can be affected by systemic biases. Here we identify such a bias, which is caused by differences in the number of library doublings, which are often not quantified in pooled chemical genomics experiments due to logistical constraints. We show that these biases, which can also potentially arise from non-specific bacterial death (e.g., when adding a drug) or from drug-induced morphological changes affecting optical density measurements, result in spurious correlations between the phenotypes of genes and drugs that can limit the interpretation of these data. Our proposed correction does not require additional information or changes to the experimental/analytical workflow, can account for any sources of difference in the number of cell doublings, and its application can drastically improve the quality of chemical genomics data, allowing this method to reach its true potential.

## Methods

**Reagents and tools table**

| Reagent/resource | Reference or source | Identifier or catalog number |
| --- | --- | --- |
| **Software** | | |
| R v4.3.3 | https://www.r-project.org | |
| gplots R package v3.1.3.1 | https://cran.r-project.org/web/packages/gplots/ | |
| Our correction code | https://github.com/horiatodor/EEL_correction | |

## Data retrieval and processing

### Li et al, 2022 data (M. tuberculosis, CRISPRi)

Per-sgRNA count data was retrieved from https://github.com/rock-lab/CGI_nature_micro_2022. Replicate counts were summed, normalized to 100,000,000 reads, and a pseudocount of 1 was added. Condition-specific changes in fitness L2FC were computed using the appropriate reference condition (from the MAGeCK filenames). Reference L2FC were estimated for 1-day, 5-day, and 10-day depletions using the two-part linear model from (Bosch

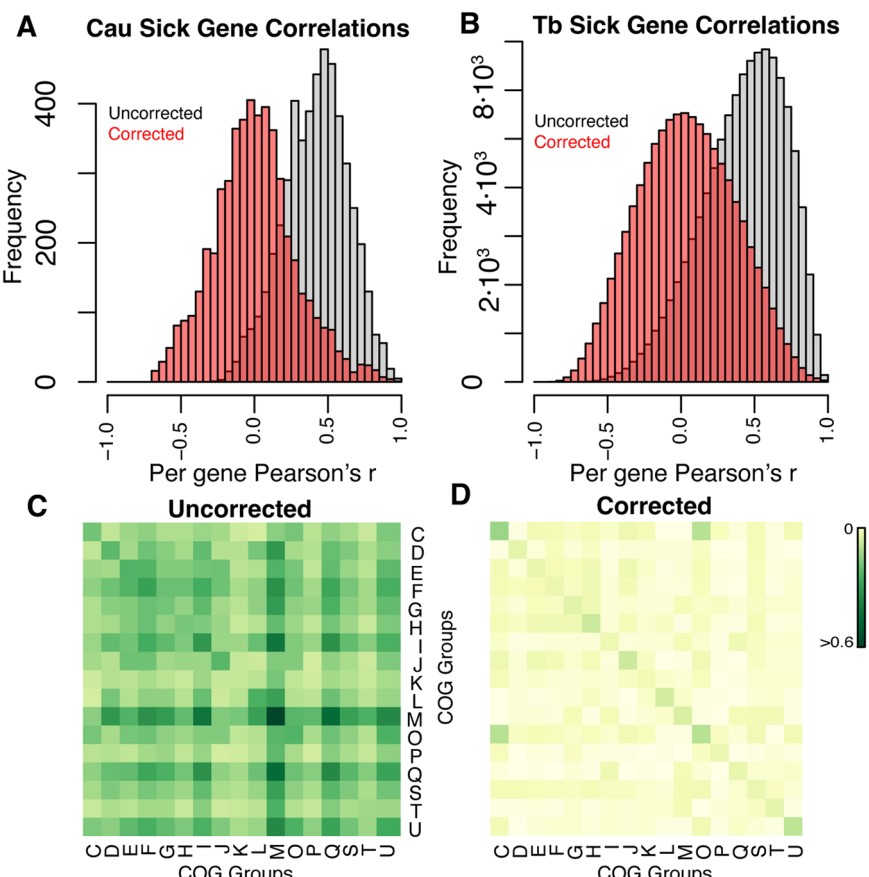

**Figure 4. Post hoc correction reduces spurious gene–gene correlations.**

(**A**, **B**) Histogram of correlation scores between all growth-deficient strains (L2FC < −1) in the *M. tuberculosis* CRISPRi dataset (*n* = 81 conditions) and the *C. crescentus* Tn-seq dataset (*n* = 198 conditions) calculated before and after correction. Consistent with the null hypothesis that most gene pairs do not have correlated phenotypes, the corrected data (red histogram) is centered around 0, with far fewer strong positive correlations than the uncorrected data (black histogram). (**C**, **D**) Heatmaps of the proportion of strong (*r* > 0.7) interactions between growth-deficient strains in COG groups in the *M. tuberculosis* CRISPRi dataset, before (**C**) and after (**D**) correcting for the number of cell doublings. Only COGs with >10 genes are included. The plethora of interactions between COGs "C" (Energy production and conversion) and "O" (Post-translational modification, protein turnover, chaperone functions) is due to the presence of cytochrome C assembly chaperones such as *ccsA* (*RVBD0529*), *ctaB* (*RVBD1451*), and *RVBD1456c*. These genes are annotated "O" but correlate with cytochrome genes ("C").

et al, 2021), since start counts for these experiments were not available. Correction was done using a bin size of 1000. To simplify analysis of the ~100,000 sgRNAs, if there were fewer than 10 sgRNAs targeting a gene, we averaged the L2FC for all those sgRNAs. If there were 10 or more sgRNAs targeting a gene, each sgRNA was assigned to one of three bins (low: 0–0.33, medium: 0.33–0.66, or high: 0.66–1) based on its predicted knockdown strength (Bosch et al, 2021). These averaged values were converted to S-scores and used for gene–gene and condition–condition correlation analyses.

### Rishi et al, 2020 data (E. coli, CRISPRi)
Per-sgRNA L2FC data was retrieved from https://github.com/hsrishi/HT-CRISPRi. Replicates were averaged. Reference fitness was the 10gen timepoint (as the dataset used was a time series, rather than a chemical screen), and correction was done using a bin size of 200.

### Price et al, 2018 data (C. crescentus and multiple other species, Tn-seq or KO)
Per-gene L2FC data was retrieved in December 2023 from https://fit.genomics.lbl.gov/. Price et al separate their data coarsely into "carbon sources", "nitrogen sources", and "stress" experiments; reference fitness was calculated as the median fitness for each gene in all "stress" experiments. Correction of *C. crescentus* Tn-seq and *E. coli* KO data was done using a bin size of 50. These values were converted to S-scores (see below) and used for gene–gene and condition–condition correlation analyses.

### Ward et al, 2024 data (A. baumannii, CRISPRi)
Per-sgRNA L2FC data was retrieved from Table S6 of (Ward et al, 2024). Reference fitness was the no-drug fitness for the relevant (t1 or t2) timepoint; we used no-drug fitness rather than median fitness because the small number of conditions could result in an inaccurate median for highly sensitive genes.

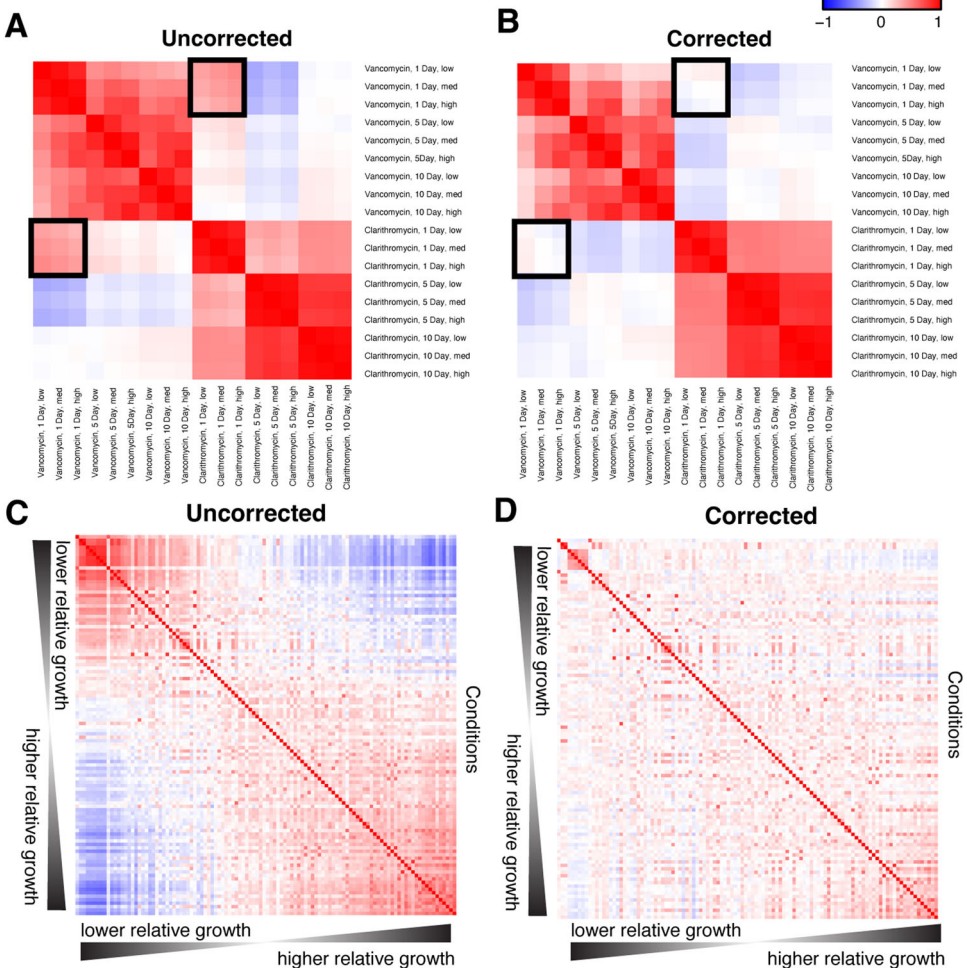

**Figure 5. Post hoc correction reduces spurious condition–condition correlations.**

(A, B) Heatmaps of correlations between vancomycin and clarithromycin treatments in *M. tuberculosis*, before and after correction. A spurious correlation between Day 1 vancomycin and clarithromycin experiments in the uncorrected data is boxed. (C, D) Correlation heatmaps of all stress experiments in *C. crescentus* before (C) and after (D) correction. Experiments are ordered by the estimated number of cell doublings. In the uncorrected data, short and long experiments show strong correlations despite limited similarity in the stresses used. This is not observed in the corrected data.

### Santiago et al, 2018 data (S. aureus, Tn-seq)

Per-gene count data was retrieved from the Supplementary Data of (Santiago et al, 2018). L2FC was calculated from "Treatment Read Counts" to the mean values column of "Control Read Counts". We focused our analysis on the 56 conditions with a median absolute deviation (MAD) < 0.5. Reference fitness was the median fitness for each gene in these conditions. Correction was done using a bin size of 50.

### Sher et al, 2020 data (C. glutamicum, Tn-seq)

Per-gene fitness data was retrieved from the Fig. 2—source data 1 file of (Sher et al, 2020). Reference fitness was calculated as the median fitness for each gene in all conditions. Correction was done using a bin size of 50.

### S-Score calculation

S-scores were calculated thus: first, we calculated the median fitness of each strain across all conditions and its dispersion about this

mean using the median absolute deviation (MAD) multiplied by 1.4826 (to align with the standard deviation). Consistent with previous work (Collins et al, 2006), a floor was applied to the MADs: all per-gene MADs below the median MAD were replaced with the median. Each gene was then normalized by subtracting its median fitness and scaled by the modified MAD. These values were used for all gene–gene and condition–condition correlation analyses. Heatmaps were generated using the heatmap.2 function from the gplots package (https://cran.r-project.org/web/packages/gplots/). No scaling was used in the generation of the heatmaps.

### Library doublings correction

To correct for biases in the number of library doublings, we used a custom in-house code written in R and freely available at https://github.com/horiatodor/EEL_correction. Briefly, the code considers all reference and condition-specific changes in fitness for a sgRNA or Tn-seq library. Genes/sgRNAs are binned based on their

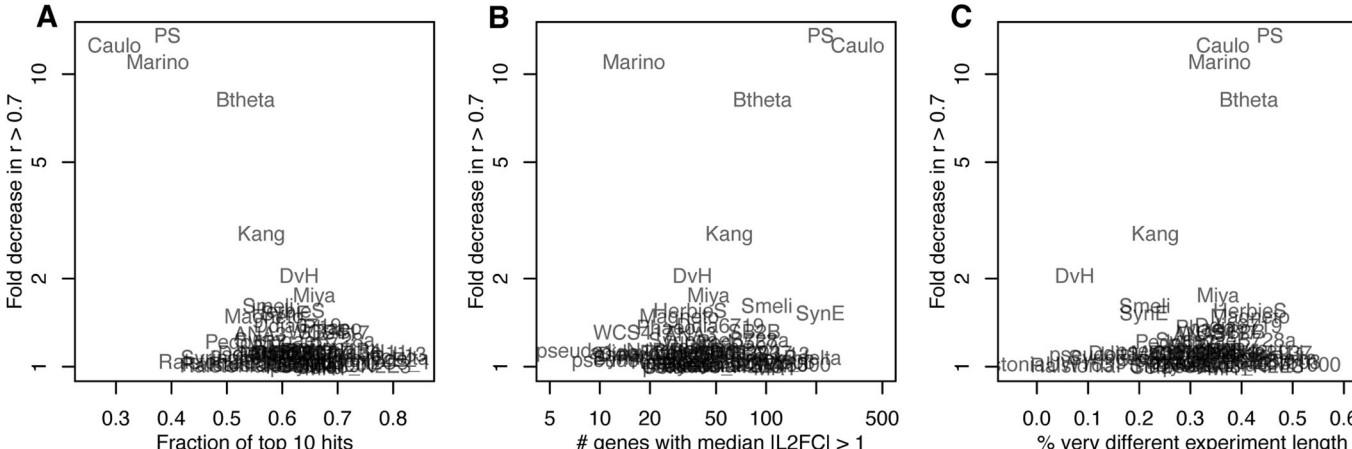

**Figure 6. The importance of correcting for differences in the number of library doublings depends on the experiment design.**

Correction was applied to all chemical genomics datasets in (Price et al, 2018). **(A)** Datasets that do not preserve their top ten covarying genes after the correction lose more of their strong correlations overall. **(B)** Datasets with more strains that are significantly more or less fit than controls (|reference L2FC| > 1) lose more strong correlations after the correction. **(C)** Datasets with more experiments requiring a larger correction (e.g., have many more or less library doublings than the reference, quantified by percent of experiments with slope of reference- vs condition-specific change in fitness more than 1 MAD away from the median) lost more strong correlations after the correction.

reference fitness. Empirically, a bin size of ~1% of the library size works well. The results of correction around the range of 1% bin size are largely equivalent (Appendix Fig. S1). Practically, for Tn-seq data, we recommend a bin size of ~50 genes, whereas for CRISPRi data larger bins (e.g., 200, 1000) can be used because there are typically more growth-deficient strains. For each bin of similarly growth-deficient strains, the median condition-specific change in fitness is calculated. This value is then subtracted from the condition-specific change in fitness of each sgRNA/gene, bringing the median of the bin to zero.

## Data availability

This study includes no data deposited in external repositories.

The source data of this paper are collected in the following database record: biostudies:S-SCDT-10_1038-S44320-024-00069-y.

## Peer review information

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

## Acknowledgements

The authors thank T Donohue, K Myers, J Peters, R Ward, and members of the CAG laboratory for extensive helpful discussions. The authors thank H Burkhart for her help with data retrieval. This work was supported by the National Institutes of Health grant R35 GM118061 (to CAG).

## Author contributions

**Lili M Kim**: Conceptualization; Software; Investigation; Visualization; Methodology; Writing—original draft; Writing—review and editing. **Horia Todor**: Conceptualization; Software; Investigation; Visualization; Methodology; Writing—original draft; Writing—review and editing. **Carol A Gross**: Supervision; Funding acquisition; Writing—original draft; Writing—review and editing.

Source data underlying figure panels in this paper may have individual authorship assigned. Where available, figure panel/source data authorship is listed in the following database record: biostudies:S-SCDT-10_1038-S44320-024-00069-y.

## Disclosure and competing interests statement

The authors declare no competing interests.

# Expanded View Figures

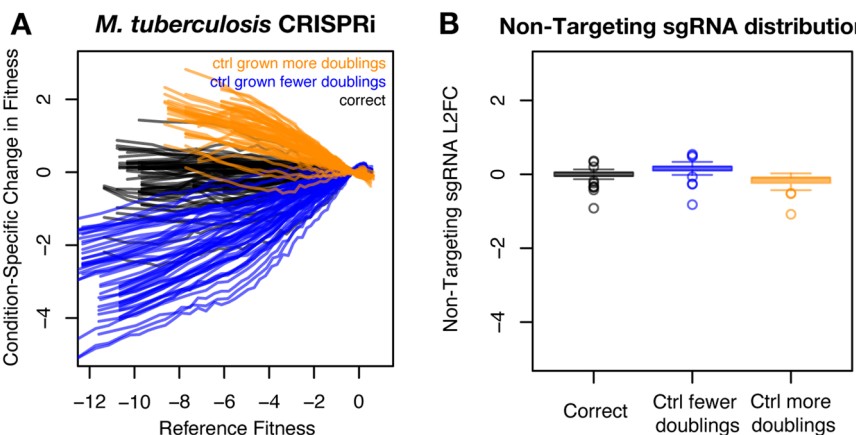

**Figure EV1.  Using deliberately incorrect controls induces correlations between the reference and condition-specific fitnesses in the *M. tuberculosis CRISPRi* data, but does not affect the non-targeting control sgRNAs.**

(A) Differences in the number of library doublings between the experimental and control conditions results in positive or negative correlations between reference and condition-specific fitnesses. For all drugs, we compared each timepoint (1, 5, or 10 days pre depletion) to a control with fewer (blue lines), the same (black lines), or more (orange lines) cell doublings (Methods, Table EV1). The incorrect comparisons resulted in strong correlations between reference and condition-specific change in fitness. If the control was grown for more cell doublings (e.g., longer pre-depletion, orange lines), sick strains would appear protected (positive condition-specific change in fitness) since they would be less depleted in the treated sample. If the control was grown for fewer cell doublings (e.g., shorter pre-depletion, blue lines), sick strains would appear sensitized (negative condition-specific change in fitness) since they would be more depleted in the treated sample. (B) The condition-specific change in fitness of non-targeting sgRNAs is not strongly affected by using non-matched control samples. The median of the non-targeting controls: correct comparison = 0.00, control with fewer doublings = 0.16, control with more doublings = -0.16.

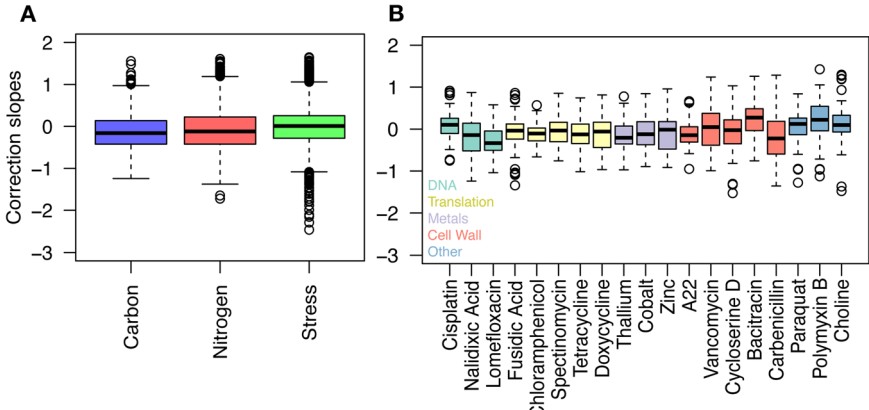

**Figure EV2. Strong negative- or positive relationships between reference and condition-specific fitnesses are not drug-specific.**

For each experiment in (Price et al, 2018), we calculated the slope of the bin median reference and condition-specific fitnesses. A slope of zero indicates a perfectly matched experiment. Neither broad categories of stresses (**A**) nor specific drugs (**B**) resulted in consistently low or high slopes, suggesting that nonzero relationships between reference and condition-specific fitnesses are due to experimental vagaries, not biological effects.

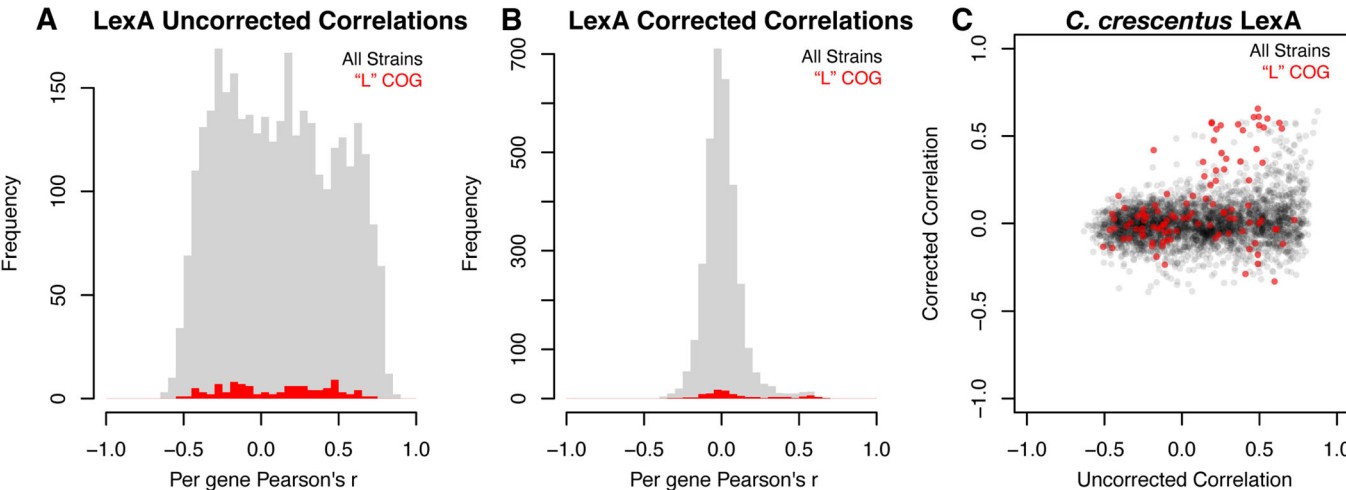

**Figure EV3. Correction improves the biological significance of strain-strain correlation with *C. crescentus* LexA (CCNA_01979).**

(A, B) Heatmap of phenotypic correlations between the *C. crescentus lexA* mutant and all other strains in the uncorrected (A) and corrected (B) datasets. Correlations depicted in red are to genes in the "L - DNA repair and recombination" COG category. (C) The same data depicted as an XY-plot, showing that "L - DNA repair and recombination" COG category genes are enriched in the strong correlations of the corrected (but not uncorrected) dataset.

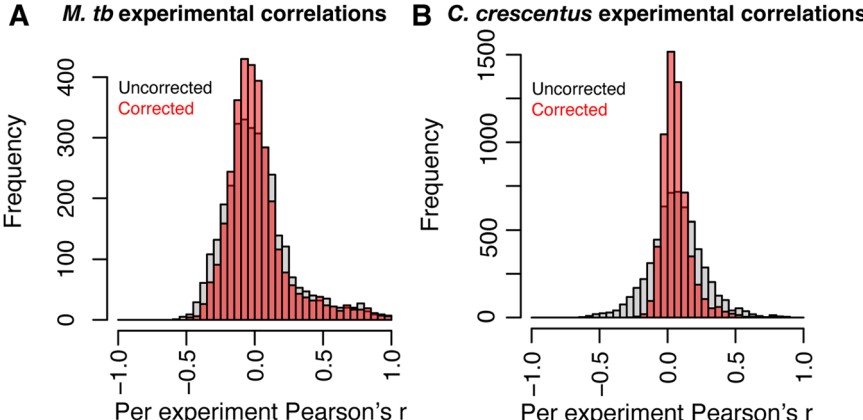

**Figure EV4. Correction removes spurious positive and negative correlations between conditions.**

(A) Histogram of correlations between 81 conditions corrected (red) and uncorrected (gray) for library doubling biases in the *M. tuberculosis* CRISPRi dataset. This dataset contains a higher proportion of same-drug experiments with high correlations to each other, and these high correlations are preserved by the correction. (B) Histogram of correlations between 198 conditions corrected (red) and uncorrected (gray) for library doubling biases in the *C. crescentus* Tn-seq dataset.

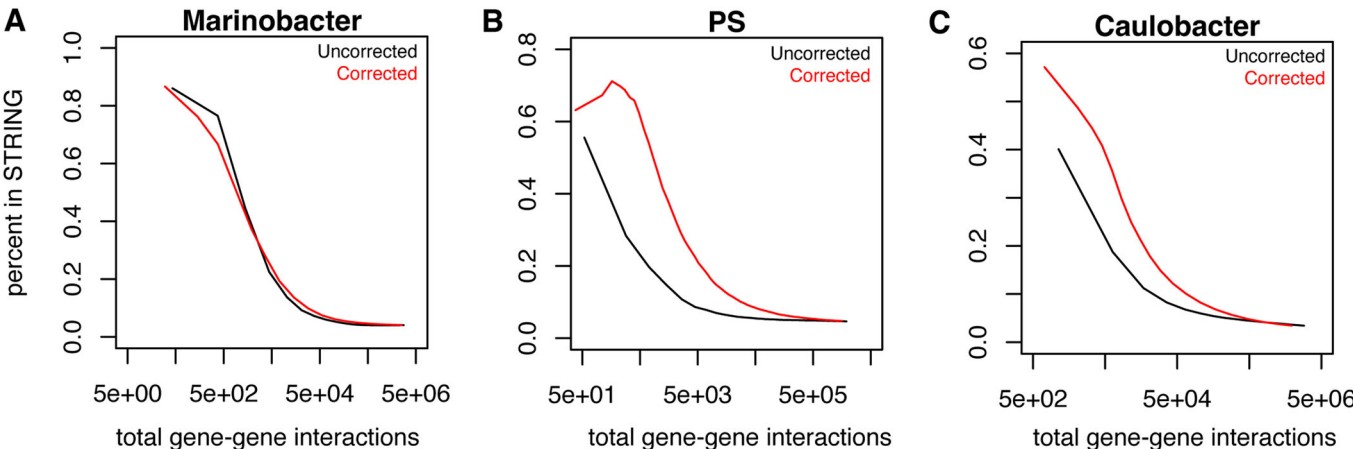

**Figure EV5.  Curves showing how the fraction of gene–gene interactions in STRING and the number of gene pairs meeting that threshold varies as the correlation threshold is changed.**

The data reveals that the correction generally improves recall of known interactions from the STRING database for diverse organisms. (**A**) *Marinobacter adhaerens HP15* (a gamma-proteobacteria). (**B**) *Dechlorosoma suillum PS* (a beta-proteobacteria). (**C**) *C. crescentus* (an alpha-proteobacteria).

