## [Peer Review File · Molecular Systems Biology]

Correction of a widespread bias in pooled chemical genomics screens improves their interpretability

Lili Kim, Horia Todor, and Carol Gross

Corresponding author(s): Horia Todor (horia.todor@ucsf.edu)

Review Timeline:

Submission Date:	30th May 24
Editorial Decision:	11th Jul 24
Revision Received:	23rd Aug 24
Editorial Decision:	13th Sep 24
Revision Received:	16th Sep 24
Accepted:	17th Sep 24

Editor: Jingyi Hou

Transaction Report:

11th Jul 2024

Manuscript Number: MSB-2024-12456

Title: Correction of a widespread bias in pooled chemical genomics screens improves their interpretability

Author: Lili Kim

Horia Todor

Carol Gross

Dear Horia,

Thank you for submitting your work to Molecular Systems Biology. I would like to apologize for the slow process, which was due to the late arrival of reviewers' reports. We have now heard back from two of the three reviewers who agreed to evaluate your manuscript. Unfortunately, we did not obtain a report from Reviewer #2. In the interest of time, and since the recommendations of the other two reviewers are quite similar, I prefer to make a decision now rather than further delay the process. You will see from the comments below that Reviewers #1 and #3 find the manuscript potentially interesting. They raise, however, several important points, which we would ask you to address in a revision of this work.

I think the reviewers' recommendations are relatively straightforward, so there is no need to reiterate their comments. Importantly, both Reviewers #1 and #3 mentioned the need for a comparison between the current method and analysis pipelines that use non-targeting controls. We would ask you to address this carefully. All other issues raised by the reviewers need to be satisfactorily addressed as well. As you may already know, our editorial policy allows in principle a single round of major revision, and it is therefore essential to provide responses to the reviewers' comments that are as complete as possible. Please feel free to contact me in case you would like to discuss in further detail any of the issues raised by the reviewers.

On a more editorial level, we would ask you to address the following issues:

- Please provide a .docx formatted version of the manuscript text (including legends for main figures, EV figures and tables). Please make sure that the changes are highlighted to be clearly visible.
- Please provide individual production quality figure files as .eps, .tif, .jpg (one file per figure).
- Please provide a .docx formatted letter INCLUDING the reviewers' reports and your detailed point-by-point responses to their comments. As part of the EMBO Press transparent editorial process, the point-by-point response is part of the Review Process File (RPF), which will be published alongside your paper.
- Please note that all corresponding authors are required to supply an ORCID ID for their name upon submission of a revised manuscript.
- We replaced Supplementary Information with Expanded View (EV) Figures and Tables that are collapsible/expandable online (see examples in <http://msb.embopress.org/content/11/6/812>). A maximum of 5 EV Figures can be typeset. EV Figures should be cited as 'Figure EV1, Figure EV2' etc... in the text and their respective legends should be included in the main text after the legends of regular figures.

Additional Tables/Datasets should be labeled and referred to as Table EV1, Dataset EV1, etc. Legends have to be provided in a separate tab in case of .xls files. Alternatively, the legend can be supplied as a separate text file (README) and zipped together with the Table/Dataset file.

For the figures and tables that you do NOT wish to display as Expanded View figures, they should be bundled together with their legends in a single PDF file called *Appendix*, which should start with a short Table of Content. Each legend should be below the corresponding Figure/Table in the Appendix. Appendix figures and tables should be referred to in the main text as: "Appendix Figure S1, Appendix Figure S2, Appendix Table S1" etc. See detailed instructions regarding expanded view here: <https://www.embopress.org/page/journal/17444292/authorguide#expandedview>.

- Before submitting your revision, primary datasets (and computer code, where appropriate) produced in this study need to be deposited in an appropriate public database (see <http://msb.embopress.org/authorguide> - dataavailability <https://www.embopress.org/page/journal/17444292/authorguide#dataavailability>).

The accession numbers and database should be listed in a formal "Data Availability" section (placed after Materials & Method) that follows the model below (see also <https://www.embopress.org/page/journal/17444292/authorguide#dataavailability>). Please

note that the Data Availability Section is restricted to new primary data that are part of this study.

Data availability

-At EMBO Press we ask authors to provide source data for the main figures. Our source data coordinator will contact you to discuss which figure panels we would need source data for and will also provide you with helpful tips on how to upload and organize the files.

- Our journal encourages inclusion of *data citations in the reference list* to directly cite datasets that were re-used and obtained from public databases. Data citations in the article text are distinct from normal bibliographical citations and should directly link to the database records from which the data can be accessed. In the main text, data citations are formatted as follows: "Data ref: Smith et al, 2001". In the Reference list, data citations must be labeled with "[DATASET]". A data reference must provide the database name, accession number/identifiers and a resolvable link to the landing page from which the data can be accessed at the end of the reference. Further instructions are available at .

- We updated our journal's competing interests policy in January 2022 and request authors to consider both actual and perceived competing interests. Please review the policy <https://www.embopress.org/competing-interests> and update your competing interests if necessary.

Please use the heading "Disclosure statement and competing interests".

- All Materials and Methods need to be described in the main text using our 'Structured Methods' format, which is required for all research articles. According to this format, the Methods section includes a Reagents and Tools Table (listing key reagents, experimental models, software and relevant equipment and including their sources and relevant identifiers) followed by a Methods and Protocols section describing the methods using a step-by-step protocol format. The aim is to facilitate adoption of the methodologies across labs. More information on how to adhere to this format as well as a downloadable template (.docx) for the Reagents and Tools Table can be found in our author guidelines: <https://www.embopress.org/page/journal/17444292/authorguide#structuredmethods>.

An example of a Method paper with Structured Methods can be found here: <https://www.embopress.org/doi/10.15252/msb.20178071>.

-Regarding data quantification:

Please ensure to specify the name of the statistical test used to generate error bars and P values, the number (n) of independent experiments (please specify technical or biological replicates) underlying each data point and the test used to calculate p-values in each figure legend. Discussion of statistical methodology can be reported in the materials and methods section, but figure legends should contain a basic description of n, P and the test applied.

Graphs must include a description of the bars and the error bars (s.d., s.e.m.).

- Please provide a "standfirst text" summarizing the study in one or two sentences (approximately 250 characters, including space), three to four "bullet points" highlighting the main findings and a "synopsis image" (550px width and 400-600 px height, PNG format) to highlight the paper on our homepage.

Here are a couple of examples:

<https://www.embopress.org/doi/10.15252/msb.20199356>

<https://www.embopress.org/doi/10.15252/msb.20209475>

<https://www.embopress.org/doi/10.15252/msb.209495>

When you resubmit your manuscript, please download our CHECKLIST (<https://www.embopress.org/pb-assets/embosite/EMBO%20Press%20Author%20Checklist-1642513524327.xlsx>) and include the completed form in your submission.

Please note that the Author Checklist will be published alongside the paper as part of the transparent process (<https://www.embopress.org/page/journal/17444292/authorguide#transparentprocess>).

If you feel you can satisfactorily deal with these points and those listed by the referees, you may wish to submit a revised version of your manuscript. Please attach a covering letter giving details of the way in which you have handled each of the points raised by the referees. A revised manuscript will be once again subject to review and you probably understand that we can give you no

guarantee at this stage that the eventual outcome will be favorable.

I look forward to receiving your revised manuscript soon.

Kind regards,
Jingyi

Jingyi Hou, PhD
Scientific Editor
Molecular Systems Biology

We realize that it is difficult to revise to a specific deadline. In the interest of protecting the conceptual advance provided by the work, we recommend a revision within 3 months (9th Oct 2024). Please discuss the revision progress ahead of this time with the editor if you require more time to complete the revisions. Use the link below to submit your revision:

IMPORTANT: When you send your revision, we will require the following items:

1. the manuscript text in LaTeX, RTF or MS Word format
2. a letter with a detailed description of the changes made in response to the referees. Please specify clearly the exact places in the text (pages and paragraphs) where each change has been made in response to each specific comment given
3. three to four 'bullet points' highlighting the main findings of your study
4. a short 'blurb' text summarizing in two sentences the study (max. 250 characters)
5. a 'thumbnail image' (550px width and max 400px height, Illustrator, PowerPoint or jpeg format), which can be used as 'visual title' for the synopsis section of your paper.

6. Please include an author contributions statement after the Acknowledgements section (see

<https://www.embopress.org/page/journal/17444292/authorguide>)

7. Please complete the CHECKLIST available at (<https://bit.ly/EMBOPressAuthorChecklist>).

Please note that the Author Checklist will be published alongside the paper as part of the transparent process

(<https://www.embopress.org/page/journal/17444292/authorguide#transparentprocess>).

See also figure legend guidelines: <https://www.embopress.org/page/journal/17444292/authorguide#figureformat>

9. Please note that corresponding authors are required to supply an ORCID ID for their name upon submission of a revised manuscript (EMBO Press signed a joint statement to encourage ORCID adoption).

(<https://www.embopress.org/page/journal/17444292/authorguide#editorialprocess>)

Currently, our records indicate that there is no ORCID associated with your account.

Please click the link below to provide an ORCID:

Link Not Available

11. Include a Reagents and Tools Table as part of the Methods section, which can be downloaded from our author guidelines (<https://www.embopress.org/page/journal/17444292/authorguide#structuredmethods>)

*** PLEASE NOTE *** As part of the EMBO Press transparent editorial process initiative (see our Editorial at <https://dx.doi.org/10.1038/msb.2010.72>), Molecular Systems Biology publishes online a Review Process File with each accepted manuscripts. This file will be published in conjunction with your paper and will include the anonymous referee reports, your point-by-point response and all pertinent correspondence relating to the manuscript. If you do NOT want this File to be published, please inform the editorial office at msb@embo.org within 14 days upon receipt of the present letter.

Reviewer #1:

Kim, Todor, and Gross present a computational method to correct for bias in chemical genomic screens. This improvement corrects for differences in the number of cell doublings that occur in the reference vs drug treatment conditions, a deviation that is common in many chemical genomic screens but is not commonly accounted for experimentally. This deviation can lead to spurious correlations in the screen output. The authors convincingly show that application of this correction factor reduces this spurious correlation and thereby improves data interpretability, particularly for chemical genomic screens that include mutants with much lower fitness than the bulk population.

Major Comments

1. Please explain in more detail how the condition-specific changes in fitness are calculated in Figure 2. Are these based on the median L2FC in bins, as described later? Are these LOESS fits?
2. One advantage of CRISPRi is that you can create non-targeting sgRNAs, which serve as internal negative controls. While the authors point out that existing analysis pipelines do not explicitly control for differences in cell doubling between references and experimental datasets, it would seem that non-targeting sgRNAs should in principle be able to be used for such normalization. For example, MAGeCK accepts flags that normalize read counts and models the null-distribution using the behavior of non-targeting control sgRNAs, as used in PMID: 35637331. Would this not achieve comparable results to the corrections the authors suggest, at least when testing conditions for significant differences? It would be useful for the authors to comment on this, and why it would or would not address this issue.
3. Related to point 2, it would be interesting to look at the behavior of non-targeting control sgRNAs (if available in a dataset) to check whether they show evidence of differences cell doubling between references and experimental datasets.
4. The authors match the pre-depletion days of the CRISPRi data from PMID: 35637331 et al. and show the effects of the correction. Another way to test their correction would be to see what happens if you intentionally compare DMSO and drug conditions from different days of pre-depletion (e.g., DMSO Day1 vs Drug Day5). One would expect comparing different days of pre-depletion intentionally would lead to spurious results (due to differences in cell doublings being important as the authors say). These spurious results should then be fixed or alleviated by the correction (e.g., results are closer to what to see when comparing the same days of pre-depletion).

Minor Comments

1. This is not necessary at all, but a recommendation would be to use some sort of non-linear or "moving" regression (like LOESS) to fit the relationship between reference fitness and condition specific effects, and using this to correct the values. This would apply a more smoothed, interpolated, corrections over the relationship, instead of the more step-wise looking corrections you get from the binning approach.

Reviewer #3:

Kim et al. identified a common artifact present in large-scale pooled screens, which comes from the uncorrected differences in cell doublings between growth conditions. This artifact may contribute to spurious calling of hits in chemical-genomics screens and the authors provided a post-hoc method for removing these effects. They used this correction on 2 published datasets, and they provided evidence that this correction improved data interpretability and uncovered novel biological insights.

The introduction effectively sets the stage for the problem the authors are addressing in the paper. In large-scale pooled experiments, condition-specific changes are usually calculated by comparing the log₂ fold-change (LFC) of the test condition to that of the reference condition. Differences in cell growth (doublings) affect the gRNA representation in the pool, impacting LFC, and therefore it crucial to account for differences in cell doublings across conditions.

A few comments for the authors to consider:

1. The authors looked at six different pooled chemical genomics datasets which used different strains and techniques. It is commendable that the authors are considering multiple datasets, highlighting the method's broad applicability. However, the reference fitness calculation they used varied for each dataset, and the bin sizes also differed. For instance, they used either a bin size of 200 or 1000 with the CRISPRi datasets. Variations in these parameters can introduce inconsistencies that affect the interpretability of the comparisons. Moreover, it would be beneficial to see an evaluation of different bin sizes, as this could significantly affect the normalization process (methodology).
2. Next, they calculated the correlation between the reference fitness and condition-specific changes in fitness for each experiment in a chemical genomics dataset. They claimed a strong relationship between the two fitness values in many experiments. The plots provided in figure 1 showed that only a fraction of the dataset exhibited this behaviour. It would be useful to provide the fraction of experiments showing this correlation and take a deeper look at the conditions for these specific cases.

While this observation doesn't negate their hypothesis, it suggests that this relationship might also be condition specific. It emphasizes the importance of correcting for differences in doublings, especially in chemical-genomics screens, where drug concentration significantly impacts doublings and thus identification of hits under different conditions.

3. The correction method was applied to two datasets, CRISPRi and Tn-seq. For both datasets, the correction method significantly decreased the strength of correlation observed between two genes across conditions, supporting the authors' claim that their pipeline removes non-specific interactions (figure 4). The authors mentioned that the correction affects genes proportional to their growth defect/advantage, and therefore most strongly impacts the relative fitness of essential genes. As such, their correction method plays an important role in understanding the relationship between essential genes as well as gene-drug interactions. The authors might consider tempering this conclusion - essential genes tend to exhibit clear fitness defects and are typically depleted from the pool. The doubling correction does not necessarily impact the identification of essential genes. Instead, it primarily serves to clean up noise levels, allowing for more confident identification of hits.

4. They also examined the correlation between drugs post correction, and they demonstrated that their method reduced the non-specific interactions between day 1 vancomycin and clarithromycin experiments in the *M. tuberculosis* CRISPRi data (figure 5). They concluded that the absence of correlation post correction suggests that it is unlikely to be biologically relevant. However, although the antibiotics have different cellular targets, the initial cellular response to stress might be similar, explaining the observed correlation. Additionally, their correction method introduced a slight negative correlation between clarithromycin day 1 and the vancomycin samples (figure 5 A-B), which has not been addressed.

5. The paper lacked a benchmarking section - comparison to published methods would strengthen the validity of their conclusion and provide context for their method's performance. A simple comparison would be correcting for differences in doublings using non-targeting controls, which is commonly used in pooled chemical CRISPR screens in mammalian cells (for example DrugZ).

Minor comments:

1. Missing references in introduction - paragraph 1, last line.
2. Missing method section for generation of heatmaps - figure 4 and 5

Summary

We would like to thank both reviewers for their thoughtful and positive reviews of our manuscript. Motivated by their suggestions, we rewrote certain sections of the text and implemented several new analyses, which we feel greatly strengthened and clarified our manuscript.

- We added a paragraph and analysis describing how non-targeting sgRNAs CANNOT be used to correct for differences in the number of cell doublings.
- We implemented the analysis suggested by reviewer #1 in which each drug condition from the *M. tuberculosis* dataset was compared to controls that were grown for fewer cell doublings or for more cell doublings (Figure EV1). As expected, this induced strong correlations between reference fitness and condition-specific changes in fitness, but did not affect the non-targeting sgRNAs.
- We analyzed how bin size affected the correction (Figure S1). We found that bin size did not strongly affect the results of the correction, and added a default bin size (1% of the total library size) to our code.
- We performed an analysis to see if any conditions were particularly likely to cause a correlation between reference fitness and condition-specific changes in fitness (Figure EV2). We did not find any conditions that systematically caused this artifact, strengthening our assertion that this is experiment specific.

Changes in the revised manuscript are in red font, and detailed responses to all reviewer comments are below.

Reviewer #1

Kim, Todor, and Gross present a computational method to correct for bias in chemical genomic screens. This improvement corrects for differences in the number of cell doublings that occur in the reference vs drug treatment conditions, a deviation that is common in many chemical genomic screens but is not commonly accounted for experimentally. This deviation can lead to spurious correlations in the screen output. The authors convincingly show that application of this correction factor reduces this spurious correlation and thereby improves data interpretability, particularly for chemical genomic screens that include mutants with much lower fitness than the bulk population.

We thank the reviewer for their positive feedback and helpful suggestions.

Major Comments

1. Please explain in more detail how the condition-specific changes in fitness are calculated in Figure 2. Are these based on the median L2FC in bins, as described later? Are these LOESS fits?

The conditions-specific changes in fitness were calculated using the median L2FC in bins, consistent with the rest of the paper.

We have added a sentence to the Figure 2 legend (Line 456-457) explaining this: “Median values of reference and condition-specific changes in fitness are calculated on a per bin basis as described (Methods) and plotted.”

2. One advantage of CRISPRi is that you can create non-targeting sgRNAs, which serve as internal negative controls. While the authors point out that existing analysis pipelines do not explicitly control for differences in cell doubling between references and experimental datasets, it would seem that non-targeting sgRNAs should in principle be able to be used for such normalization. For example, MAGeCK accepts flags that normalize read counts and models the null-distribution using the behavior of non-targeting control sgRNAs, as used in PMID: 35637331. Would this not achieve comparable results to the corrections the authors suggest, at least when testing conditions for significant differences? It would be useful for the authors to comment on this, and why it would or would not address this issue.

The relative abundance of non-targeting controls does not change as a function of time since these strains do not have a growth defect. Indeed, the L2FC of all strains is frequently adjusted so that the median of the non-targeting controls is 0 (e.g. Hawkins et al., 2020 *Cell Systems*). As such, the non-targeting controls cannot be used to infer (or control for) experiment length.

To demonstrate this, we implemented Reviewer #1’s idea (major point #4) and generated inherently biased L2FC data by comparing each of the drug conditions from (Li et al., 2022) to DMSO (control) samples grown for fewer cell doublings or for more cell doublings than appropriate.

As expected, mismatched control/experimental pairs resulted in a strong correlation between reference fitness and condition-specific changes in fitness. The slope of the correlation depended on whether the control was collected after growing for more cell doublings or for fewer cell doublings than the experimental sample. Importantly, only extremely minor changes to the median L2FC of the non-targeting control sgRNAs were caused by the use of incorrect controls, underscoring the impossibility of using non-targeting controls to correct for the number of cell doublings. The figure (EV1) is reproduced below.

We added a paragraph to the first section of the results highlighting the impossibility of using non-targeting sgRNAs to control for the number of cell doublings and describing the analysis above (Line 96-112). We also added Figure EV1, above, illustrating this analysis.

3. Related to point 2, it would be interesting to look at the behavior of non-targeting control sgRNAs (if available in a dataset) to check whether they show evidence of differences cell doubling between references and experimental datasets.

Non-targeting sgRNAs do not (and should not) exhibit any effects from uncorrected differences in the number of cell doublings. Please see the response to point 2 above for a more detailed explanation.

4. The authors match the pre-depletion days of the CRISPRi data from PMID: 35637331 et al. and show the effects of the correction. Another way to test their correction would be to see what happens if you intentionally compare DMSO and drug conditions from different days of pre-depletion (e.g., DMSO Day1 vs Drug Day5). One would expect comparing different days of pre-depletion intentionally would lead to spurious results (due to differences in cell doublings being important as the authors say). These spurious results should then be fixed or alleviated by the correction (e.g., results are closer to what to see when comparing the same days of pre-depletion).

We'd like to thank the reviewer for this excellent suggestion. We implemented this analysis and found that intentionally comparing different days of pre-depletion clearly demonstrates the importance of the number of cell doublings on the relationship between reference and condition-specific changes in fitness. This analysis has been added to the text (Lines 96-110). Please see the response to point #1 above for additional details.

Minor Comments

1. This is not necessary at all, but a recommendation would be to use some sort of non-linear or "moving" regression (like LOESS) to fit the relationship between reference fitness and condition specific effects, and using this to correct the values. This would apply a more smoothed, interpolated, corrections over the relationship, instead of the more step-wise looking corrections you get from the binning approach.

We appreciate this suggestion, and strongly considered this while designing our correction. Ultimately, we settled on using bins. Our reasoning is that the most important corrections come at the far end of the reference fitness distributions - i.e. the least fit genes - where LOESS and similar approaches often exhibit unconstrained and unpredictable behavior. Because of this, we chose to stick with the binning strategy, which, while not perfect, is at least easy to understand and predictable.

Reviewer #3

Kim et al. identified a common artifact present in large-scale pooled screens, which comes from the uncorrected differences in cell doublings between growth conditions. This artifact may contribute to spurious calling of hits in chemical-genomics screens and the authors provided a post-hoc method for removing these effects. They used this correction on 2 published datasets, and they provided evidence that this correction improved data interpretability and uncovered novel biological insights.

The introduction effectively sets the stage for the problem the authors are addressing in the paper. In large-scale pooled experiments, condition-specific changes are usually calculated by comparing the log₂ fold-change (LFC) of the test condition to that of the reference condition. Differences in cell growth (doublings) affect the gRNA representation in the pool, impacting LFC, and therefore it crucial to account for differences in cell doublings across conditions.

We thank the reviewer for their positive feedback and thoughtful suggestions.

A few comments for the authors to consider:

1. The authors looked at six different pooled chemical genomics datasets which used different strains and techniques. It is commendable that the authors are considering multiple datasets, highlighting the method's broad applicability. However, the reference fitness calculation they used varied for each dataset, and the bin sizes also differed. For instance, they used either a bin size of 200 or 1000 with the CRISPRi datasets. Variations in these parameters can introduce inconsistencies that affect the interpretability of the comparisons. Moreover, it would be beneficial to see an evaluation of different bin sizes, as this could significantly affect the normalization process (methodology).

We thank the reviewer for these excellent suggestions.

The reference fitness was generally calculated as the median L2FC across all conditions. There were two chemical genomics experiments where the fitness was calculated differently. In both cases the difference was due to experimental exigencies. For the *M. tuberculosis* experiments (Li et al., 2022), the reference fitness was calculated based on a previous paper (Bosch et al., 2021) because only endpoint samples were collected and sequenced, so a true reference fitness could not be calculated. For the *A. baumannii* CRISPRi experiments (Ward et al., 2023), the no-drug fitness was used because there were only a few conditions and we wanted to avoid inaccurate medians for highly sensitive genes (e.g. OM genes). We have added a description of these considerations to the relevant sections in Methods (Lines 278-279, 283-286, 292-294).

Regarding bin size, we have empirically found that a bin size of ~1% of the total number of strains in the experiment produces good results. Moreover, in the range around 1%, the results are largely invariant to the bin size. We have added this information to the methods, set a default bin size in the code of 1% (Lines 323-325), and added an additional figure (Appendix Figure S1, reproduced below) showing that the correction is bin size invariant around 1%.

2. Next, they calculated the correlation between the reference fitness and condition-specific changes in fitness for each experiment in a chemical genomics dataset. They claimed a strong relationship between the two fitness values in many experiments. The plots provided in figure 1 showed that only a fraction of the dataset exhibited this behaviour.

A majority of the experiments exhibited a non-zero slope when comparing relative and reference fitness. For example, 77/81 of the *M. tuberculosis* experiments have a significant ($p < 0.001$) relationship between binned reference and condition-specific changes in fitness. Similarly, 4301/6966 of the Tn-seq experiments have a significant ($p < 0.001$) relationship between binned reference and condition-specific changes in fitness.

We believe that the confusion was caused by highlighting only a few experiments in red and green in Figure 2 and in the main text sentence describing the figure. As such, we updated the Figure 2 legend to make it clearer that we only highlighted the most extreme cases (Line 462-464). It now reads:

“Many experiments exhibited significant correlations between relative and condition-specific changes in fitness (i.e. the line is not flat). Experiments with exceptionally positive or negative slopes are highlighted in green and red (respectively).”

Additionally, we removed the references to the green and red lines from the main text (Line 130). That sentence now reads:

“In almost all data sets, we found **significant** relationships between reference fitness and condition-specific changes in fitness in many experiments (Figure 2), both positive and negative, suggesting uncorrected differences in the number of cell doublings between experiments.”

It would be useful to provide the fraction of experiments showing this correlation and take a deeper look at the conditions for these specific cases. While this observation doesn't negate their hypothesis, it suggests that this relationship might also be condition specific. It emphasizes the importance of correcting for differences in doublings, especially in chemical-genomics screens, where drug concentration significantly impacts doublings and thus identification of hits under different conditions.

The initial version of our manuscript highlighted that some clarithromycin experiments in the *M. tb* dataset had positive slopes while others had negative slopes, which suggests that the observed slope is caused by experimental vagaries, not biological effects. We agree with the reviewer that drug concentration can have a large impact on doublings; however, the extent and consistency of that impact within drug categories depends on how the experiments were done.

To determine more broadly if certain conditions predispose bacteria to too many or too few cell doublings, we considered the Tn-seq data from (Price et al., 2018), which includes >6,000 experiments in 40+ bacteria. Price et al. separate their data coarsely into “carbon sources”, “nitrogen sources”, and “stress” experiments. All three categories included positive and negative relationships between reference to condition-specific changes in fitness in approximately equal proportions (A, below). A more focused look at 18 common conditions revealed a similar result (B, below). As such, we do not believe changes in the number of cell doublings are condition-specific. Rather, we believe they arise largely from experimental vagaries independent of specific conditions. We have added this figure (as EV2) and analysis to the text (Line 133-136):

“In several cases, we observed both positive and negative relationships between reference fitness and condition-specific changes in fitness for different concentrations of the same drug (e.g. Figure 2A, purple lines), and we found no conditions or drugs that consistently caused a high or low number of cell doublings (Figure EV2).”

3. The correction method was applied to two datasets, CRISPRi and Tn-seq. For both datasets, the correction method significantly decreased the strength of correlation observed between two genes across conditions, supporting the authors' claim that their pipeline removes non-specific interactions (figure 4). The authors mentioned that the correction affects genes proportional to their growth defect/advantage, and therefore most strongly impacts the relative fitness of essential genes. As such, their correction method plays an important role in understanding the relationship between essential genes as well as gene-drug interactions. The authors might consider tempering this conclusion - essential genes tend to exhibit clear fitness defects and are typically depleted from the pool. The doubling correction does not necessarily impact the identification of essential genes. Instead, it primarily serves to clean up noise levels, allowing for more confident identification of hits.

We thank the reviewer for this suggestion and we agree that our correction is not required for identifying essential genes within a genome. However, chemical genomics screens do probe how modulating essential gene expression (for example, via mismatches in the sgRNA) affects susceptibility to antibiotics, and how essential genes interact with other genes in a pathway (correlation being a proxy for this).

An example of this can be found in the manuscript describing the *M.tb* CRISPRi dataset (Li et al., 2022). A key finding is that knockdown of essential genes involved in mycolic acid synthesis, specifically sensitizes cells to rifampicin, vancomycin, and bedaquiline. Our correction preserves these real relationships while discarding spurious ones, which, as the reviewer notes, allows for the more confident identification of this interaction. As such, we would prefer to leave the text unaltered.

4. They also examined the correlation between drugs post correction, and they demonstrated that their method reduced the non-specific interactions between day 1 vancomycin and clarithromycin experiments in the *M. tuberculosis* CRISPRi data (figure 5). They concluded that the absence of correlation post correction suggests that it is unlikely to be biologically relevant. However, although the antibiotics have different cellular targets, the initial cellular response to stress might be similar, explaining the observed correlation. Additionally, their correction method introduced a slight negative correlation between clarithromycin day 1 and the vancomycin samples (figure 5 A-B), which has not been addressed.

In Fig 5A, uncorrected data, the average boxed correlation between vancomycin and clarithromycin is 0.38; this correlation is abolished by our correction, as we highlight. However, the reviewer rightly points out that there is a negative correlation induced by our correction between clarithromycin day 1 and vancomycin samples. This induced correlation is of substantially lower magnitude than the correlation removed by our correction: the average values are -0.11 and -0.15.

We have changed the manuscript in two ways to address this issue. First, we changed the color scaling in Figure 5 to be continuous rather than 9 discrete colors, giving a more accurate visual impression of the correlation strength. Second, we added a panel to Figure EV4 to include a histogram of all *M. tb* experimental correlations before and after the correction. This histogram shows that our correction results in fewer strong positive or negative correlations overall. We added a reference to this in the main text (Line 192-193): “It also abolished other likely spurious correlations (Figure EV4A).”

5. The paper lacked a benchmarking section - comparison to published methods would strengthen the validity of their conclusion and provide context for their method's performance.

Our correction is presented as an addendum to common analysis packages, rather than in place of these analyses. Because of this, we structured the paper to benchmark analyzed data corrected with our code against analyzed data which was not, and show that our correction removes spurious correlations and improves the quality and usability of data. As such, we prefer to leave the text unaltered.

A simple comparison would be correcting for differences in doublings using non-targeting controls, which is commonly used in pooled chemical CRISPR screens in mammalian cells (for example DrugZ).

Non-targeting sgRNAs cannot be used to control for differences in doublings, as their relative abundance does not change as the experiment progresses. For more details, see our response to Point #2 of Reviewer #1's comments.

Minor comments:

1. Missing references in introduction - paragraph 1, last line.

We added references to six chemical genomics papers in diverse bacteria (Nichols et al. 2011; Santiago et al. 2018; Price et al. 2018; Sher, Lim, and Bernhardt 2020; Li et al. 2022; Ward et al. 2024) (Line 47-48).

2. Missing method section for generation of heatmaps - figure 4 and 5

We added a description of how the heatmaps were generated in the Methods (Line 315-316). It reads:

“Heatmaps were generated in R using the heatmap.2 function from the gplots package (<https://cran.r-project.org/web/packages/gplots/>). No scaling was used in the generation of the heatmaps.”

13th Sep 2024

Manuscript Number: MSB-2024-12456R

Title: Correction of a widespread bias in pooled chemical genomics screens improves their interpretability

Author: Lili Kim

Horia Todor

Carol Gross

Dear Horia,

Thank you for sending us your revised manuscript. We have now received the feedback from Reviewer #1 who has agreed to re-evaluate your work. As you will see below, the reviewer is satisfied with the modifications made. Due to the unavailability of Reviewer #3, we asked Reviewer #1 to also assess your responses to Reviewer #3's comments. Reviewer #1 found your responses to be reasonable and has recommended publication.

Before we can formally accept your manuscript, please address the following editorial level issues:

1. Please provide up to five keywords in the manuscript file.
2. Please remove the "Authors' contribution" sections from the manuscript file.
3. Please ensure that Figures 2B-F are called out.
4. Please upload the "Reagents and Tools Table" as a separate .docx file. The template can be found in our author guidelines: <https://www.embopress.org/page/journal/17444292/authorguide#structuredmethods>.
5. Please add a "Data availability" section. Since this study does not generate large-scale datasets, please include only the following sentence in this section - "This study includes no data deposited in external repositories".
6. Table EV1 should be uploaded as a single Excel file with the legend included either above the table or as a separate tab, rather than in a zip folder.

When you resubmit your manuscript, please download our CHECKLIST (<https://bit.ly/EMBOPressAuthorChecklist>) and include the completed form in your submission. *Please note* that the Author Checklist will be published alongside the paper as part of the transparent process (<https://www.embopress.org/page/journal/17444292/authorguide#transparentprocess>)

Click on the link below to submit your revised paper.

Kind regards,
Jingyi

Jingyi Hou, PhD
Scientific Editor
Molecular Systems Biology

If you do choose to resubmit, please click on the link below to submit the revision online before 13th Oct 2024.

IMPORTANT: When you send your revision, we will require the following items:

1. the manuscript text in LaTeX, RTF or MS Word format
2. a letter with a detailed description of the changes made in response to the referees. Please specify clearly the exact places in the text (pages and paragraphs) where each change has been made in response to each specific comment given
3. three to four 'bullet points' highlighting the main findings of your study
4. a short 'blurb' text summarizing in two sentences the study (max. 250 characters)
5. a 'thumbnail image' (550px width and max 400px height, Illustrator, PowerPoint or jpeg format), which can be used as 'visual title' for the synopsis section of your paper.
6. Please include an author contributions statement after the Acknowledgements section (see <https://www.embopress.org/page/journal/17444292/authorguide#manuscriptpreparation>)
7. Please complete the CHECKLIST available at (<https://bit.ly/EMBOPressAuthorChecklist>). Please note that the Author Checklist will be published alongside the paper as part of the transparent process (<https://www.embopress.org/page/journal/17444292/authorguide#transparentprocess>).
8. When assembling figures, please refer to our figure preparation guideline in order to ensure proper formatting and readability in print as well as on screen:
<https://bit.ly/EMBOPressFigurePreparationGuideline>
See also figure legend guidelines: <https://www.embopress.org/page/journal/17444292/authorguide#figureformat>
9. Please note that corresponding authors are required to supply an ORCID ID for their name upon submission of a revised manuscript (EMBO Press signed a joint statement to encourage ORCID adoption). (<https://www.embopress.org/page/journal/17444292/authorguide#editorialprocess>)
Currently, our records indicate that the ORCID for your account is 0000-0001-5556-6085.

Please click the link below to modify this ORCID:
Link Not Available

10. Include a Reagents and Tools Table as part of the Methods section, which can be downloaded from our author guidelines (<https://www.embopress.org/page/journal/17444292/authorguide#structuredmethods>)

*** PLEASE NOTE *** As part of the EMBO Press transparent editorial process initiative (see our Editorial at <https://dx.doi.org/10.1038/msb.2010.72> , Molecular Systems Biology will publish online a Review Process File to accompany accepted manuscripts. When preparing your letter of response, please be aware that in the event of acceptance, your cover letter/point-by-point document will be included as part of this File, which will be available to the scientific community. More information about this initiative is available in our Instructions to Authors. If you have any questions about this initiative, please contact the editorial office (msb@embo.org).

Reviewer #1:

The authors have addressed all of my comments.

Title: Correction of a widespread bias in pooled chemical genomics screens improves their interpretability

Response to editorial and reviewer comments

Author responses are marked in red.

Editorial

1. Please provide up to five keywords in the manuscript file.

Keywords have been added after the abstract.

2. Please remove the "Authors' contribution" sections from the manuscript file.

The Author Contribution section has been removed.

3. Please ensure that Figures 2B-F are called out.

Thank you for pointing this out. Figures 2B-F are now included in the callout in line 131.

4. Please upload the "Reagents and Tools Table" as a separate .docx file. The template can be found in our author

guidelines: https://www.embopress.org/page/journal/17444292/authorguide#structured_methods.

We have included a separate .docx file with the Reagents and Tools Table and removed the embedded table from the revised manuscript.

5. Please add a "Data availability" section. Since this study does not generate large-scale datasets, please include only the following sentence in this section - "This study includes no data deposited in external repositories".

A Data Availability section has been added after the Materials and Methods section.

6. Table EV1 should be uploaded as a single Excel file with the legend included either above the table or as a separate tab, rather than in a zip folder.

We have added a legend as a separate tab in the Table EV1 Excel file and discarded the zip folder.

Reviewer

Reviewer #1:

The authors have addressed all of my comments.

We thank the reviewers for their time and expertise.

17th Sep 2024

Manuscript number: MSB-2024-12456RR

Title: Correction of a widespread bias in pooled chemical genomics screens improves their interpretability

Dear Horia,

Thank you again for sending us your revised manuscript. We are now satisfied with the modifications made and I am pleased to inform you that your paper has been accepted for publication.

Kind regards,
Jingyi

Jingyi Hou, PhD
Scientific Editor
Molecular Systems Biology
